# Putative Roles and Therapeutic Potential of the Chaperone System in Amyotrophic Lateral Sclerosis and Multiple Sclerosis

**DOI:** 10.3390/cells13030217

**Published:** 2024-01-24

**Authors:** Leila Noori, Vahid Saqagandomabadi, Valentina Di Felice, Sabrina David, Celeste Caruso Bavisotto, Fabio Bucchieri, Francesco Cappello, Everly Conway de Macario, Alberto J. L. Macario, Federica Scalia

**Affiliations:** 1Department of Biomedicine, Neuroscience and Advanced Diagnostics (BIND), University of Palermo, 90127 Palermo, Italy; leili.noori.1359@gmail.com (L.N.); vahid.saqagandomabadi@you.unipa.it (V.S.); valentina.difelice@unipa.it (V.D.F.); sabrina.david@unipa.it (S.D.); celeste.carusobavisotto@unipa.it (C.C.B.); fabio.bucchieri@unipa.it (F.B.); francesco.cappello@unipa.it (F.C.); 2Department of Anatomy, School of Medicine, Medical University of Babol, Babol 47176-47745, Iran; 3Euro-Mediterranean Institute of Science and Technology (IEMEST), 90139 Palermo, Italy; econwaydemacario@som.umaryland.edu (E.C.d.M.); ajlmacario@som.umaryland.edu (A.J.L.M.); 4Department of Microbiology and Immunology, School of Medicine, University of Maryland at Baltimore—Institute of Marine and Environmental Technology (IMET), Baltimore, MD 21202, USA

**Keywords:** neurodegenerative diseases, amyotrophic lateral sclerosis, multiple sclerosis, chaperone system, protein misfolding, protein homeostasis, chaperonopathy, chaperonotherapy

## Abstract

The putative pathogenic roles and therapeutic potential of the chaperone system (CS) in amyotrophic lateral sclerosis (ALS) and multiple sclerosis (MS) are reviewed to provide a bibliographic and conceptual platform for launching research on the diagnostic and therapeutic applications of CS components. Various studies suggest that dysfunction of the CS contributes to the pathogenesis of ALS and MS, and here, we identify some of the implicated CS members. The physiology and pathophysiology of the CS members can be properly understood if they are studied or experimentally or clinically manipulated for diagnostic or therapeutic purposes, bearing in mind that they belong to a physiological system with multiple interacting and dynamic components, widespread throughout the body, intra- and extracellularly. Molecular chaperones, some called heat shock protein (Hsp), are the chief components of the CS, whose canonical functions are cytoprotective. However, abnormal chaperones can be etiopathogenic factors in a wide range of disorders, chaperonopathies, including ALS and MS, according to the data reviewed. Chaperones typically form teams, and these build functional networks to maintain protein homeostasis, the canonical role of the CS. However, members of the CS also display non-canonical functions unrelated to protein homeostasis. Therefore, chaperones and other members of the CS, if abnormal, may disturb not only protein synthesis, maturation, and migration but also other physiological processes. Thus, in elucidating the role of CS components in ALS and MS, one must look at protein homeostasis abnormalities and beyond, following the clues emerging from the works discussed here.

## 1. Introduction

Amyotrophic lateral sclerosis (ALS) [1] and multiple sclerosis (MS) [2,3] are serious diseases that affect the brain and the spinal cord with deleterious impacts on skeletal muscles throughout the body. They share many symptoms; in fact, misdiagnosis may occur and may influence the treatment options, disease prognosis, and patient’s quality of life; however, they are distinct diseases. For both, there exist some disease-modifying therapies, particularly for MS, but there is no curative therapy for them. Given this and their high incidence, especially in the young population, attention should be paid to the molecules involved and novel therapeutic strategies, as pointed out in this review. The role of members of the chaperone system (CS) in ALS and MS is not fully understood despite clinical and experimental data gathered over the years, indicating that they could be involved in the pathogenesis of these disorders. The data indicate that CS remodeling in neuronal tissue occurs in these two disorders, whose characterization could help in the identification of CS components with pathophysiological functions peculiar to one or the other. This is a brief review of these data that was prompted by the cogent need for a platform for launching the search and development of novel strategies and means for efficaciously preventing and treating ALS and MS, which, at the present time have bad prognosis and no satisfactory therapy.

The CS is a physiological system composed of molecular chaperones; chaperone co-factors; co-chaperones; and chaperone receptors and interactors [4]. The chief components of the CS are the molecular chaperones, some of which are called heat shock proteins (Hsps). All the members of the CS can be classified considering molecular weight, which provides a panoramic view of the field useful for clinicians and pathologists in practice and research. Thus, the following groups have been delineated in kDa: over 200; 100–199; 81–99; 65–80; 55–64; 35–54; and 34 or less. Within these ranges, there are families of phylogenetically related chaperones, for example, the Hsp90, Hsp70, Hsp60 (the chaperonins), and sHsp (small Hsp) families [5]. 

Although the canonical functions of the CS are typically cytoprotective and are directed to maintain protein homeostasis under physiological normal conditions and in response to stress, some of its members, if abnormal in concentration, structure, or location, can be etiopathogenic factors and cause diseases, chaperonopathies [6], and those that particularly affect the nervous system are grouped as neurochaperonopathies [7]. The identification of a chaperone as a pathogenic factor opens the road toward developing and applying therapeutic means to target it, namely, chaperonotherapy. This can be positive or negative, with the former aimed at boosting a chaperone or replacing it when defective or absent. In contrast, negative chaperonotherapy consists of removing or blocking–inhibiting a malfunctioning chaperone that causes pathologic lesions [8].

## 2. Amyotrophic Lateral Sclerosis (ALS)

Neuromuscular diseases (NMDs) include various neurodegenerative disorders that primarily affect the peripheral nervous system and the skeletal muscles, causing clinical manifestations, such as muscle weakness and atrophy, and abnormalities of tendons and distal sensory function [9]. ALS, also called motor neuron disease (MND) or Lou Gehrig’s disease [10], is a catastrophic disorder affecting the upper and lower motor neurons in the cortex, brainstem, and spinal cord, resulting in progressive paralysis [11]. Most ALS cases are sporadic (sALS), and hereditary familial types (fALS) are less common [11]. Genetic abnormalities in ALS, including some pertaining to the CS [12], are under scrutiny. Mutations in the *TAR DNA-binding protein 43* (*TDP-43*), *C9orf72*, and *FUS* genes are associated with the accumulation of aggregated proteins in motor neurons in ALS [13,14]. 

Mutations in the *valosin-containing protein* (*VCP*) gene are involved in fALS and sALS cases [15]. VCP has chaperoning functions, controlling protein degradation during cell stress. Impaired VCP is associated with an excess of misfolded proteins, endoplasmic reticulum (ER) stress, and activated autophagy in ALS [16]. Sequestration of aberrant proteins into inclusions is a common feature of ALS and other neurodegenerative disorders that can be characterized as proteinopathies [17]. 

Also, mutations in the gene encoding the chaperone *Ubiquilin 2* (*UBQLN2*) are associated with X-linked forms of ALS [18]; this age-related form of ALS shows dysfunctional mitophagy caused by altered function of the UBQLN2 protein, which interacts with Hsp70 [19].

Another causative gene associated with ALS is *Sigma receptor 1* (*SIGMAR1*, *SigR1*), a ligand-activated chaperone protein (improperly defined as a “receptor“, as it was mistaken for an opioid receptor) that resides in the mitochondrion-associated endoplasmic reticulum membrane (MAM) of the cell [20,21]. Variants in *SIGMAR1* are associated with ALS, frontotemporal lobar degeneration (FTLD), and motor neuron disease (MND) in humans [22,23]. 

The structure of SIGMAR1 is unique and performs a variety of functions. For instance, it is involved in gene transcription and lipid biosynthesis; it regulates ion channels, receptors, and kinases; and it controls oxidative stress, enhancing the nuclear production of antioxidant proteins during ER stress [21]. Its translocation ability, i.e., moving through the ER, MAM, and nuclear membrane, allows it to function more like a signaling modulator between organelles than a canonical receptor. A typical chaperone activity performed by SIGMAR1 occurs during the folding of nascent proteins in the ER lumen. This function is strictly related to its MAM localization, in which it acts as a bridge between organelles, allowing for the maintenance of mitochondrial integrity and functions, which, in turn, regulate cell survival and death and ROS production. 

SIGMAR1 is capable of binding various intracellular proteins but is usually bound to the binding immunoglobulin protein (BiP), another element of the chaperone system involved in protein folding and protein quality control. The presence of specific ligands or, more commonly, a decrease in intraluminal calcium levels determines the detachment of SIGMAR1 from the BiP and its intracellular activation [21]. 

The effects of *SIGMAR1* mutations have been studied in vitro and in vivo. *SIGMAR1*-mutated NSC34 cells show abnormal subcellular distribution of the SIGMAR1 protein and intracellular protein aggregation, and NSC34 cells with *SIGMAR1* knocked out suffer ER dysfunction with associated autophagy and vacuolization [24]. Pharmacological activation of SIGMAR1 via an agonist reverts the neurotoxic effect in NSC34 cells [24]. In motor neurons extracted from *Sigmar1*−/− knockout mouse embryos or in motor neurons treated with the SIGMAR1 antagonist, morphofunctional axonal degeneration occurred, but the sensory neurons and glial cells treated with the antagonist were not damaged, demonstrating their lower sensitivity to SIGMAR1 dysfunction and confirming the cytotype selectivity of the mutation, which has also been observed in humans ALS [25].

*Drosophila* expressing the human S1R mutant shows an ocular morphological defect and reduced motor abilities [26]. Furthermore, mitochondrial fragmentation is also characteristic of mutated fly tissues [26]. Similarly, in *Sigmar1*−/− mice, neuromuscular dysfunction has been confirmed to be associated with mitochondrial and ER damage [25]. These data together prove the neuroprotective power of SIGMAR1. The therapeutic potential that this CS element may have for ALS treatment is also suggested by results from the zebrafish *SIGMAR1*-ALS model, treated with an agonist of the receptor [27]. This would be applicable not only to ALS but also in general for the management of neurodegenerative disorders [28].

Variants of the *superoxide dismutase1* (*SOD1*) gene have been implicated in fALS, and its protein product is one of the proteins commonly aggregated in patient tissues [29]. Intracellular aggregates associated with mutant SOD1 cause intracellular toxicity and stimulate neuronal cell death [30]. 

Specifically with regard to protein homeostasis in ALS, cases associated with the *SOD1* mutant are accompanied by dysfunction of the CS (Figure 1). Genes and proteins compromised in the presence of mutant *SOD1* are involved in protein degradation; the immunological response; cell death/survival; the heat shock response, involving Hsp70 genes such as *HSPA1B* and *HSPA4*; and include Hsp22 (*HSPB8*) and anti-apoptotic protein BCL2-associated athanogene 3 (Bag3), which increases the activity of BCL2 and also interacts with the HSC70/Hsp70 proteins [31].

Mutant SOD1 is accompanied by dysregulation of a subgroup of chaperones, including a decrease in CCT chaperonin subunits 5 and 6Aζ; Hsp70; and Hsp40 and increase in Hsp25 and Hsp27 [12]. A considerable increase in Hsp27 was observed in mutant *SOD1* N2a mouse neuroblastoma cells after a mild heat shock, which protected these cells against ensuing cell death [32]. In mice overexpressing Hsp27 and bearing the SOD1^G93A^ mutation, the heat shock protein has a positive effect on diseased motor neurons; however, this beneficial effect occurred only in the first stages of the experimental disease [33]. Protein aggregates contain sequestered Hsps in the presence of mutant SOD1 in motor neurons in human and mouse models of ALS [34]. Aggregates of mutant SOD1 colocalize with Hsp27, Hsp40, and Hsp70 and decrease their functions in motor neurons [35,36].

The diminished efficiency of the CS results in the failure of protection against the proteotoxicity of protein aggregates in motor neurons of the spinal cord in transgenic mice overexpressing mutant SOD1 [37]; therefore, strategies to enhance the performance of chaperones have been a focus of therapeutic approaches [38]. In this regard, the evidence in the literature is visibly growing, and many works are of considerable interest. For instance, recently, it was demonstrated that synthesized constructs are able to activate and exploit the endogenous lysosomal chaperone-mediated autophagy (CMA) pathway specifically against pathogenic SOD1 aggregates, determining a decrease in aggregates and improving the outcome of ALS in vivo [39]. Activated *Hsp* gene transcription improved hind limb muscle performance, postponed the death of motor neurons, and increased the lifespan of SOD1^G93A^ transgenic mice [40]. It was found that pharmacological molecules defined as HSR amplifiers, e.g., arimoclomol, have the potential to improve the ALS phenotype of the spinal cord and brain of VCP mutant mice and SOD1^G93A^ mice [41,42]. Improvements in terms of decrease in protein aggregates and activation/expression of neuronal Hsp70 have been demonstrated in human fibroblast cells and iPSC-derived motor neurons isolated from ALS patients and treated with arimoclomol [41]. Overexpression of Hsp70 can suppress protein aggregation and apoptosis in mutant SOD1 neural cells [35].

Expression of a constitutively active version of the main heat shock transcription factor, Hsf1, and administration of Hsf1 activators such as geldanamycin can greatly increase the expression of Hsps and protect cells in a primary culture model of fALS [43,44]. In vivo overexpression of the human molecular chaperone HSJ1a (hHSJ1a) decreased mutant SOD1 aggregation, strengthened muscle force, enhanced motor neuron survival, and improved the disease in SOD1^G93A^ transgenic mice [45].

Increased levels of mitochondrial chaperonin Hsp60, Hsp70, and Hsp27 induced by *Withania somnifera* root extract (WS) in a mutant SOD1 mouse model of ALS, correlate with motor neuron survival and function and enhanced longevity [46]. On the other hand, mitochondrial damage occurs upon abnormal interaction of TDP-43 and FUS proteins with Hsp60 in mutant TDP-43 and FUS mouse models of ALS [47]. The CCT complex, a Group II chaperonin, can suppress protein aggregation in neurodegenerative conditions [48]. Elevated levels of CCT5, CCT7, and CCT8 subunits in the CCT complex occur in an in vitro model of ALS, suggesting that CCT is active in diseased cells [49]. 

Neurodegenerative diseases, including ALS, show an accumulation of misfolded proteins and aggregates thereof. The activation of molecular chaperones is a defense mechanism in these proteinopathies that prevents the formation of aggregates and dissolves those already formed, a fact that makes members of the CS attractive therapeutic targets for the chaperonotherapy of ALS [50]. The chaperones Hsp27, Hsp70, and Hsp90 and the CCT subunits are the most studied CS members regarding ALS pathogenesis. However, in vitro and in vivo results are not all in agreement about their precise roles and mechanisms of action in ALS. Serum levels of Hsp70 and Hsp90 were found to be higher in patients suffering from ALS compared to healthy controls, while Hsp27 levels did not differ from controls [51]. Further studies are necessary to elucidate whether changes in the levels of members of the CS reflect a physiological attempt to protect the human body or they are pathogenic, favoring disease initiation and/or progression.

Neuroinflammation is another hallmark of ALS pathophysiology, a mechanism underpinning demyelination. Immune system stimulation with the production of inflammatory mediators in microglia and astrocytes has been observed during ALS development associated with loss of motor neurons [52,53]. Aggregated proteins and danger-associated molecular patterns (DAMPs), including extracellular Hsps released by injured cells, are considered pro-inflammatory factors in ALS [54]. The transient receptor potential (TRP) channels recognizing DAMPs are upregulated upon oxidative stress in ALS [55,56]. Some Hsps are ligands for CD14 and TLR2 receptors, so extracellular Hsps such as HspB5 can be involved in microglial M1/M2 and astrocyte A1/A2 polarization to induce inflammatory or immunoregulatory phenotypes [57,58,59]. Typical cytoprotective Hsps can switch over to being pathogenic when they are quantitatively/qualitatively abnormal and thereby make ALS a chaperonopathy-causing neuroinflammation [53]. Molecular chaperones such as Hsp27 and αB-crystallin capture extracellular misfolded SOD1 and TDP-43 and reduce their immune triggering effect in an in vitro model of ALS [60]. Hsp27 and αB-crystallin are also neuroprotective through improved phagocytic activity of microglia in a mouse model of ALS [60,61]. The macrophage migration inhibitory factor (MIF) showed chaperone activity targeting misfolded SOD1 proteins, suggesting its induction as a potential therapeutic approach [62]. Several clinical trials were unsuccessful, in that they did not show favorable effects of anti-inflammatory medicines for ALS neuroinflammation [63]. The study of chaperonopathies, along with neuroinflammation in ALS, is an active, emerging field that offers promise for developing chaperonotherapy for this disease, particularly considering the known interactions of the CS and the immune system [64,65].

## 3. Multiple Sclerosis (MS)

Multiple sclerosis (MS) is a chronic autoimmune–inflammatory condition of the central nervous system (CNS) and the commonest non-traumatic disabling disease affecting young adults [66]. MS is characterized by axonal demyelination and gliosis, deficient sensory function, muscle spasticity and weakness, and gait abnormalities [66,67]. Clinical phenotypes of MS are linked to chronological and biological aging [68]. It has been suggested that poor treatment outcomes are associated with aging of the immune and central nervous systems [68]. Immune-mediated and neurodegenerative processes contribute to the pathophysiology of MS [69]. Different genetic, epigenetic, and environmental variables stimulate myelin-specific peripheral T lymphocytes to invade the myelin sheet during MS development [70]. One of the most recently studied environmental causes is Epstein–Barr (EB) virus infection [71,72]. However, whether neuronal damage is caused by antiviral immunity or whether the virus triggers an autoimmune response is not yet clear. In relation to this, it is interesting to highlight that, since many members of the CS are evolutionarily conserved and present in all living systems, it cannot be excluded that molecular similarity (cross-reactive immunogenic–antigenic epitopes) between human CS and viral proteins may elicit autoimmune reactions [73]. Cross-similarity as the possible cause of autoimmunity has also been reported for SARS-CoV-2 [74].

Several EB virus proteins have a certain level of similarity with human proteins [75], and some of the latter are molecular chaperones, for example, αB-crystallin, for which antibody cross-reactivity with the Epstein–Barr nuclear antigen 1 (EBNA1) viral protein has been reported [76]. Extracellular Hsps mediate intercellular signaling communicating with cells via TLR2 and TLR4, which are implicated in neurodegeneration and MS development [77,78]. As already mentioned in the above section dedicated to ALS, the αB-crystallin released by oligodendrocytes can activate microglia and the M1/M2 switch of macrophages. In this way, αB-crystallin may exert its neuroprotective and anti-inflammatory effects [52]. However, it has been reported that, in MS pathophysiology, the αB-crystallin is released in excess and it activates perivascular macrophages and B lymphocytes that act as functional antigen-presenting cells (APCs) [79]. In addition, in the human innate repertoire, HspB5-reactive memory T cells, which, if activated by a high concentration of αB-crystallin, produce INFγ and thereby subvert the response, culminating in tissue damage [80]. Viewed from this perspective, HspB5 may have a central role in MS, acting as a “fake antigen” following EBV infection and activating an abnormal immune response. MS cells express high levels of Hsps, along with inflammation and oxidative stress [81]. An abnormal increase in the levels of some molecular chaperones may trigger the release of inflammatory factors and autoimmunity, which may become etiopathogenic mechanisms in MS (Figure 2) [82]. 

Thus, it may be that immune suppression is beneficial in MS, as proposed with respect to Hsp70 [83]. 

Besides its role in maintaining protein homeostasis, including misfolded proteins, Hsp70 contributes to the immune response in MS [84]. Genes encoding Hsp70 and genes involved in antigen processing and presentation pathways are overexpressed in MS brains [83]. HSPA1A, a member of the Hsp70 family, is increased in MS lesions, suggesting that it plays a role as a pro-inflammatory factor [85]. Also, members of the Hsp70 family, such as HSPA1A, HSPA1B, and HSPA6, cooperate with members of the Hsp90 family, such as HSP90AB1, in promoting antigen presentation [86]. Hsp70 gene overexpression is accompanied by an increase in myelin autoantigen presentation [87]. Hsps are thought to interact with the major histocompatibility complex (MHC) in this process [88,89]. Components of the MHC class II, HLA-DQB1, and HLA-DRB5 genes are upregulated in different areas of the brain in MS patients [85]. Hsp70 also causes the activation of nuclear factor (NF)-κB and enhances the production of the proinflammatory cytokines IL-1β, IL-6, and TNF-α via CD14 [90]. The interaction of Hsp70 with the myelin basic protein (MBP) and proteolipid protein (PLP), the key proteins of the myelin sheath, suggests its vital function in the regenerative process in MS [91]. Hsp70 quantities are increased close to remyelinated areas, while poor remyelination tends to occur in association with a lack of Hsp70 in MS autopsies [90,92]. 

Stress may impair the ability of molecular chaperones to refold misfolded proteins, resulting in the apoptotic cascade and cell death [93]. Hsp60 has shown both pro- and anti-inflammatory roles when upregulated in MS. The former action is assisted by monocytes, B cells, and effector T cell signaling, whereas the anti-inflammatory action relies on B cells and regulatory T cells [94]. Hsp60 is augmented in response to secreted cytokines, including IL-1β, TNF-α, IL-4, IL-6, and IL-10, in adult human astrocytes [95]. Numerous roles are attributed to Hsp60 in the pathogenesis of different autoimmune diseases, e.g., diabetes and arthritis, but the mechanistic details of chaperonin immunopathogenic participation in MS are still poorly understood [81]. 

Elevated levels of Hsp60 [96] and Hsp90 have been found in the sera and cerebrospinal fluids (CSFs) of MS patients [97]. Increased Hsp90 and decreased oligodendrocyte precursor cells (OPCs) cause an autoimmune reaction and activate the complement system, disturbing remyelination in MS [98]. The gene coding for αB-crystallin (HspB5) is upregulated, and its protein product is the most abundant protein in the astrocytes and oligodendrocytes of MS plaques, but it is only sporadically detectable in demyelinated axons [99]. Anti-αB-crystallin-specific antibodies have been detected in the sera and spinal fluids of MS patients [100], and for many years, it was thought that this protein could be an autoantigen recognized by the host’s immune system; however, its overexpression is anti-inflammatory [101]. Over the years, it has been proven that αB-crystallin and other sHsps bind immunoglobulins (Igs) as the receptors of, and not antigens for, Igs [101]. Moreover, sHsps can bind Igs in a temperature-dependent manner, which can be interpreted to be a therapeutic action aimed at decreasing inflammation by capturing Igs in vivo [102]. sHsps modify the concentration of Igs in CSF [101], but the dosages used are too low to affect anti-myelin-specific Igs from sera in therapeutic trials [102]. The levels of CCT6A, a subunit of the CCT complex, are increased in plasma from patients with autoimmune diseases such as systemic lupus erythematosus (SLE) and rheumatoid arthritis (RA), in which CCT6A may represent an autoantigen recognizable by γδ T cells [103]. Cells expressing CCT6A on their surfaces are not attacked by cytotoxic T cells when treated with anti-CCT6A antibodies [103]. Accordingly, CCT6A may be considered a novel ligand for γδ T cells in human autoimmune diseases, and to explain its presence in patient plasma and on cell membrane surfaces, it has been suggested that CCT6A in the plasma membrane may be recognized by γδ T cells, leading to tissue damage, which, in turn, can release CCT6A into the plasma and sustain the activation of γδ T cells [103]. Although no data have been reported in the literature regarding the expression of the CCT6A subunit in the blood of MS patients, its possible involvement in autoimmune processes should also be explored in MS. Interestingly, on the other hand, a noticeable level of autoantibodies against CCT or Hsp60 have been observed in autoimmune diseases, but their functional significance is still not clearly understood. [104]. Immunization with full-length Hsps (e.g., Hsp60, Hsp10, Hsp40, and Hsp70) could delay the onset of autoimmune disease, as demonstrated in murine models [105,106], paving the way to possible beneficial effect of Hsp immunization. However, further research is needed to establish effective and risk-free dosages of Hsps in combination with standard therapies.

## 4. Conclusions

ALS and MS are neurological disorders affecting the nervous system that have some similarities regarding muscle and nerve damage, but they differ in etiology, genetics, symptoms, progression, and life expectancy. Another important similarity is that, although progress has been made in determining possible therapeutic targets and new approaches [107,108], there is still no cure for these diseases. 

The canonical CS functions are typically intracellular and are directed toward the maintenance of protein homeostasis, including promoting the proper folding of nascent peptides, the re-folding of partially denatured proteins, and the elimination of irreversibly denatured proteins. Aggregation of misfolded proteins in the cells occurs in neurodegenerative diseases, for example, ALS. Therefore, failure of the CS plays a critical role in the development of neurodegeneration. Molecular chaperones, the chief members of the CS, are secreted naturally or under pathophysiological and stress conditions into the extracellular environment and body fluids. The immune system recognizes extracellular chaperones as “danger signals” and as “self-adjuvants”, and thus, they can stimulate innate and acquired immune responses to produce pro-inflammatory cytokines and reinforce antigen presentation, respectively. Consequently, autoantibodies against molecular chaperones have been detected in the peripheral blood of MS patients. 

Taking into consideration that neuroinflammation involving molecular chaperones is a key pathogenic factor in some neurodegenerative diseases and that the CS participates in the pathogenesis of ALS and MS (Figure 1 and Figure 2), it is pertinent to consider chaperonotherapy a possible therapeutic approach [8].

Positive chaperonotherapy includes replacing a defective gene or chaperone protein or boosting a defective chaperone protein with docking compounds like pharmacological chaperones. Negative chaperonotherapy consists of knocking down a mutant gene coding for a pathogenic variant or blocking–inhibiting a pathogenic chaperone. Both types of chaperonotherapy ought to be investigated to determine if their potential therapeutic value for ALS and MS can be confirmed (Table 1).

In the last few years, chemical compounds, such as 4-phenylbutyrate (PBA), have emerged as promising therapeutic agents for disorders caused by misfolding or aggregation of proteins [111]. 

With regard to mutations in chaperone genes, these pharmacological agents are promising for treating chaperonopathies and other protein-misfolding diseases [112]. Missense mutations produce proteins that are structurally altered and unable to function properly, for example, mutations of chaperones that cause chaperonopathies. The pharmacological agents can bind to the mutated chaperones and stabilize them, helping them to fold correctly and regain their function. Unlike traditional therapies that target all proteins of a particular type, these pharmacological agents can be designed to specifically bind to and stabilize the mutated chaperone.

Pharmacological docking compounds specific to a mutant chaperone can also be combined with other therapeutic interventions in a synergistic approach. For example, the compound could be used in combination with gene therapy to deliver a functional copy of the mutated chaperone gene into the patient’s cells (Table 1). 

## Figures and Tables

**Figure 1 cells-13-00217-f001:**
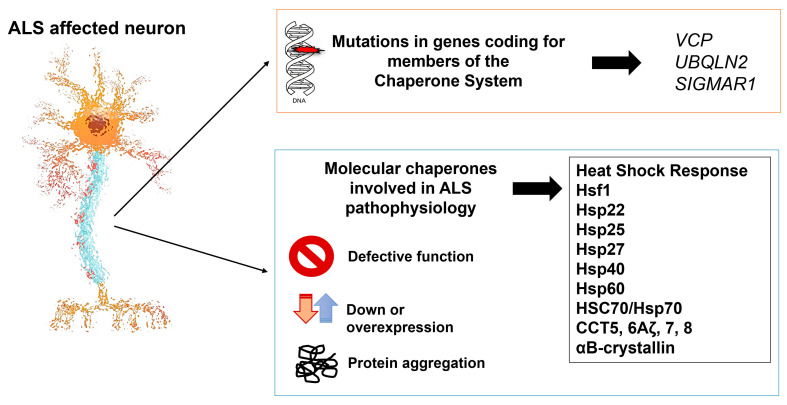
Members of the chaperone system involved in ALS. The impairment of the neurons during ALS may be caused by mutations in three gene members of the CS shown at the top right corner. Moreover, defective function, down- or overexpression, and the intra- or extracellular aggregation of several molecular chaperones (boxed at bottom right) are associated with ALS’s pathophysiology. Abbreviations: *VCP*: *valosin-containing protein*; *UBQLN2: Ubiquilin 2*; *SIGMAR1*: *SigR1*, *Sigma receptor 1*; Hsf1: heat shock transcription factor 1; Hsp: heat shock protein; CCT: chaperonin-containing tailless complex polypeptide 1.

**Figure 2 cells-13-00217-f002:**
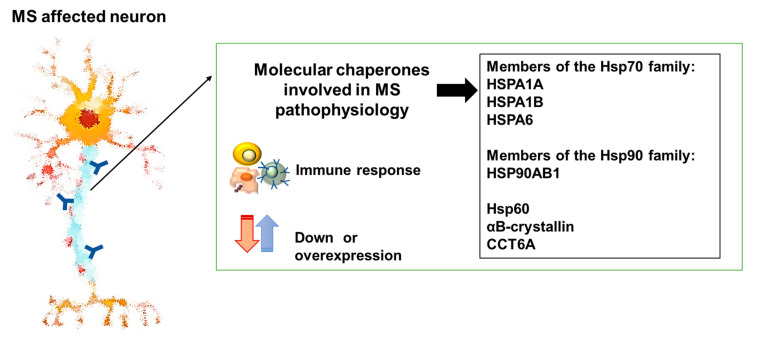
Members of the chaperone system involved in MS. Some molecular chaperones may cause an abnormal immune response in MS, and down- or overexpression of some of them is associated with the pathophysiology of MS.

**Table 1 cells-13-00217-t001:** Example of potential chaperonotherapies for ALS and MS diseases.

Molecular Chaperones	Potential Chaperonotherapy for ALS
VCP	Negative, inhibitors of VCP [109], Positive, HSR amplifiers [41,42]
SIGMAR1	Positive, agonist of SIGMAR1 [27,110]
SOD1	Positive, HSR amplifiers [41,42] or boosting specific molecular chaperones [60,61]
	**Potential Chaperonotherapy for MS**
Hsp60 Hsp70 Hsp90 sHsps	A combination of negative and positive modes of chaperonotherapy could be implemented, including inhibition of the proinflammatory Hsp60, 70, and 90 [83,84,97,98] and induction of sHsps.

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
