# Peer review of "Putative Roles and Therapeutic Potential of the Chaperone System in Amyotrophic Lateral Sclerosis and Multiple Sclerosis"

_cells, 2024, doi:10.3390/cells13030217_

Round 1

Reviewer 1 Report

Comments and Suggestions for Authors

the review is very informative in regards to the role of the chaperone system in amyotrophic lateral sclerosis and multiple sclerosis. Two minor points could be adress 

- figures could be improved, especially figure legends that are very short. More explanation are needed in addition to the drawing. Maybe a table would also help to have a complete vision of the CS in these pathologies.

- I am a little bit confused with the CCT6A part. could you clarify it ?

Author Response

We thank the Reviewers for their very useful comments and suggestions, which we have followed in preparing the revised manuscript now submitted. We added explanations and changed various parts of the text, as indicated by the Reviewers. All new or modified parts are visible in Track Changes. Below this message is a point-by-point response to the Reviewers’ comments.

We hope the manuscript is now acceptable for publication in Cells (MDPI).

Sincerely,

Federica Scalia

Reviewer 1 comment #1: the review is very informative in regards to the role of the chaperone system in amyotrophic lateral sclerosis and multiple sclerosis. Two minor points could be adress

- figures could be improved, especially figure legends that are very short. More explanation are needed in addition to the drawing. Maybe a table would also help to have a complete vision of the CS in these pathologies.

Authors’ Reply: We thank the reviewer for this comment. We modified the legends according to the suggestion.

Reviewer 1 comment #2: - I am a little bit confused with the CCT6A part. could you clarify it ?

Authors’ Reply: We thank the reviewer for this comment. We modified the text according to the suggestion.

Reviewer 2 Report

Comments and Suggestions for Authors

This is a potentially interesting minireview about chaperones in amyotrophic lateral sclerosis and multiple sclerosis, but there is room for improvement:

1) At the beginning of the Introduction the authors should explain why they have decided to group and focus on ALS and MS. This should also be mentioned in the Abstract.  Usually ALS is grouped with other neurodegenerative diseases or neuromuscular disorders.

2) Some more detail should be given in the Introduction to the description of the different families of chaperones and their mechanism of action, besides just mentioning their classification by MW. Additional chaperones should also be mentioned, such as those in subcellular organelles, for example in the ER.

3) There is some deficiency in the flow of the text. Some parts seem to be a list of disconnected short sentences each summarizing a different reference, although some of these studies are very related to each other and their description should be integrated in the text, reducing redundancies. For example how mutant SOD1 affects chaperones and their expression in page 3.

4) Mutant Sigma-1 receptor should also be mentioned as a cause of ALS in Fig. 1 and in the text.

5) When listing the names of the different chaperones in the figures, the authors should also list the synonyms, such as HspB5 and others.

6) I suggest to add a table with a list of potential chaperone mediated therapies for ALS and MS, perhaps classified by each chaperone, from reports of studies that modulate chaperone levels by overexpression, knockdown, activation or inhibition, including citations. 

Comments on the Quality of English Language

The English is good. Some minor editing is needed.

Author Response

We thank the Reviewers for their very useful comments and suggestions, which we have followed in preparing the revised manuscript now submitted. We added explanations and changed various parts of the text, as indicated by the Reviewers. All new or modified parts are visible in Track Changes. Below this message is a point-by-point response to the Reviewers’ comments.

We hope the manuscript is now acceptable for publication in Cells (MDPI).

Sincerely,

Federica Scalia

Reviewer 2 comment #1: This is a potentially interesting minireview about chaperones in amyotrophic lateral sclerosis and multiple sclerosis, but there is room for improvement:

At the beginning of the Introduction the authors should explain why they have decided to group and focus on ALS and MS. This should also be mentioned in the Abstract.  Usually ALS is grouped with other neurodegenerative diseases or neuromuscular disorders.

Authors’ Reply: We thank the reviewer for this comment. We modified the text according to the suggestion.

Reviewer 2 comment #2: Some more detail should be given in the Introduction to the description of the different families of chaperones and their mechanism of action, besides just mentioning their classification by MW. Additional chaperones should also be mentioned, such as those in subcellular organelles, for example in the ER.

Authors’ Reply: We thank the reviewer for the comment, however, we believe that this information is easily available from numerous papers in the literature and especially in the works cited in this review. A dissertation on the elements of CS (including co-chaperones) would make this work extremely extensive and would distract the reader from the main purpose of the review.

Reviewer 2 comment #3 There is some deficiency in the flow of the text. Some parts seem to be a list of disconnected short sentences each summarizing a different reference, although some of these studies are very related to each other and their description should be integrated in the text, reducing redundancies. For example how mutant SOD1 affects chaperones and their expression in page 3.

Authors’ Reply: We thank the reviewer for this comment. We modified the text according to the suggestion. How SOD1 modifies the expression of some CS elements is not known, but the increase or decrease in expression of some CS elements in the presence of SOD1 mutant has been determined and is reported in the text (line 115).

Reviewer 2 comment #4 Mutant Sigma-1 receptor should also be mentioned as a cause of ALS in Fig. 1 and in the text.

Authors’ Reply: We thank the reviewer for this comment. We modified the text and the figure according to the suggestion.

Reviewer 2 comment #5 When listing the names of the different chaperones in the figures, the authors should also list the synonyms, such as HspB5 and others.

Authors’ Reply: We thank the reviewer for this comment. We modified the figure according to the suggestion.

Reviewer 2 comment #6 I suggest to add a table with a list of potential chaperone mediated therapies for ALS and MS, perhaps classified by each chaperone, from reports of studies that modulate chaperone levels by overexpression, knockdown, activation or inhibition, including citations. Authors’ Reply: We thank the reviewer for this comment. We follow the suggestion of the reviewer and we add a Table 1.  

Reviewer 3 Report

Comments and Suggestions for Authors

The manuscript "Putative roles and therapeutic potential of the Chaperone system in amyotrophic lateral sclerosis and multiple sclerosis" concerns the important role of the chaperone system in pathogenesis and potential treatment. The role of the chaperone proteins is important for both disorders and this review is helpful, aggregating information on both disorders.

The minor comment concerns the lack of explanation what is the common point of view for these both diseases, having quite different pathomechanisms and progression. Describing two disorders only in the context of chaperones function is not enough to give the conceptual platform for therapeutic application of the chaperone system.

Author Response

We thank the Reviewers for their very useful comments and suggestions, which we have followed in preparing the revised manuscript now submitted. We added explanations and changed various parts of the text, as indicated by the Reviewers. All new or modified parts are visible in Track Changes. Below this message is a point-by-point response to the Reviewers’ comments.

We hope the manuscript is now acceptable for publication in Cells (MDPI).

Sincerely,

Federica Scalia

Reviewer 3 comment #1:

The manuscript "Putative roles and therapeutic potential of the Chaperone system in amyotrophic lateral sclerosis and multiple sclerosis" concerns the important role of the chaperone system in pathogenesis and potential treatment. The role of the chaperone proteins is important for both disorders and this review is helpful, aggregating information on both disorders.

The minor comment concerns the lack of explanation what is the common point of view for these both diseases, having quite different pathomechanisms and progression. Describing two disorders only in the context of chaperones function is not enough to give the conceptual platform for therapeutic application of the chaperone system.

Authors’ Reply: We thank the reviewer for this comment. We modified the text according to the suggestion.

Round 2

Reviewer 2 Report

Comments and Suggestions for Authors

The review is significantly improved.

Comments on the Quality of English Language

The English is good. Some minor editing is needed.

Author Response

We thank again the reviewer. The improvement of the manuscript occurred thanks to her/his comments.

Kind Regards